# OpenReview forum: "On Rademacher Complexity-based Generalization Bounds for the Transformer Architecture"
_ICLR.cc/2026/Conference — ICLR 2026 Conference Withdrawn Submission_

### Official Review · Reviewer_4Bg6 · 2025-10-18

**Soundness:** 2
**Presentation:** 1
**Contribution:** 1
**Rating:** 2
**Confidence:** 5

**Summary:**

The authors claim to propose the first end-to-end, data-dependent generalization bound for the Transformer architecture using the Rademacher complexity framework. The authors derive a theoretical measure of the model’s effective capacity by analyzing the Lipschitz continuity of the self-attention. The bound they provide depends on the - sequence length (n), embedding norms (BX), and weight norms (BW), which is reasonable under the assumptions. Using this result, they iteratively bound the Rademacher complexity for a single attention head, multi-head self-attention (MHSA), a full Transformer block (including LayerNorm and feed-forward layers), and finally a deep L-layer Transformer through inductive composition. The use of Talagrand’s Contraction Lemma allows for the bounds obtained in this work. Section 3..7 then provides the final generalization risk bound using the standard framework. A numerical check on synthetic data verifies the theoretical scaling: the bound’s magnitude increases monotonically with depth, consistent with observed model capacity growth. The paper focuses on the use of pre LayerNorm  architecture to derive these bounds.

**Strengths:**

1. The paper provides a Rademacher complexity based bound on the transformer function class, under standard/simple enough assumptions on the norm bounds. They provide a simple supporting experiment to show that the bound holds in practice, with the monotonic increase shown in figure 1.

**Weaknesses:**

1. One major criticism is the incompleteness of the Related Works section where the authors have not discussed the existing works on deriving the bounds for the Lipschitz constant of the self-attention mechanism. See [1-3] below.  Furthermore, the authors’ claim of the first work characterizing the Rademacher complexity is incorrect, see [4] below.

2. In the draft, the relevant citations are missing in all important places. Some examples – no citation for the Talagrand’s Contraction lemma, lack of appropriate references for the Rademacher Complexity and the transformers paper, etc

3. This is a very explicit claim about the writeup of the paper, but the lack of any equation number and furthermore the lack of appropriate cross referencing, lack of appropriate lemma indexing and such, is somewhat unprofessional. While this specific paper is easy to follow, it is generally good practice to do so.

4. It seems that the derivations in line 274-294 on page 6 have flipped the composition of the LayerNorm with the MHSA and FFN. Can the authors elaborate this?

5. The experimental results are too trivial and completely lacking any informative findings. Furthermore, no details of the synthetic copying task were provided in the paper.

[1] Kim, H., Papamakarios, G., & Mnih, A. (2021). The Lipschitz Constant of Self-Attention. In Proc. ICML 2021, PMLR vol. 139

[2] Castin, V., Ablin, P., & Peyré, G. (2024). How Smooth Is Attention? (ICML 2024). Proceedings of Machine Learning Research vol. 235

[3] Wang, Y., Chauhan, J., Wang, W., & Hsieh, C.-J. (2023). Universality and Limitations of Prompt Tuning. In Advances in Neural Information Processing Systems 36 (NeurIPS 2023).

[4] Trauger, J., & Tewari, A. (2024). Sequence Length Independent Norm-Based Generalization Bounds for Transformers. In Proceedings of the 27th International Conference on Artificial Intelligence and Statistics (PMLR, Vol. 238, pp. 1405–1413)

**Questions:**

1. Can the authors elaborate which specific variant of the Contraction lemma has been invoked to have the factor of 2 in line 169 of page 4 -> Rˆ S(ϕ ◦ G) ≤ 2LϕRˆ S(G) ?

2. Can the authors describe further details on the use of LLMs? I use Gemini myself, and the quality of this output is way too similar to what Gemini-2.5 Pro generates by default. No effort seems to have been made to improve the manuscript.

3. Can the authors describe further why the related works section was left in such an incomplete state?

4. Why were the experimental details left out? More importantly, the lack of experimental details raise skepticism on how the experiments were performed, even though the findings are straightforward and provided quite explicitly in the literature over the years.

---

### Official Review · Reviewer_AiMt · 2025-10-30

**Soundness:** 1
**Presentation:** 1
**Contribution:** 1
**Rating:** 2
**Confidence:** 5

**Summary:**

The paper studies the generalization bounds for Transformer architectures. The authors claim to have obtained the first theoretical analysis of the self-attention mechanism within the Rademacher complexity framework. Section 3.7 aims to give a formal generalization bound using empirical Rademacher complexity from one Transformer block. Some numerical results are shown regarding the Gen gap and the model depth.

The paper is written in collaboration with Google's Gemini language model.

**Strengths:**

Theoretical analysis of Transformer's generalization bound using the Rademacher complexity is an interesting topic.

**Weaknesses:**

The paper is written in collaboration with Google's Gemini language model. The soundness of the results needs to be verified.

1. L319 mentions a stability assumption, which I have not found in the paper.

2. What does the approximation relation mean exactly in L320?

3. The equation in L328 is invalid according to L310.

4. Is the loss function in L368 for supervised learning?

5. L392 mentions Theorem 3.6, which I'm unable to locate.

6. The description on the experimental design is vague. Details are not provided regarding the training (e.g., data, architecture, hyper-parameters...) It is unclear to me how the empirical gaps are computed.

7. The presentation needs to be improved. The main results are not formed into mathematical theorems.

**Questions:**

See Weakness

---

### Official Review · Reviewer_3CYc · 2025-10-31

**Soundness:** 2
**Presentation:** 2
**Contribution:** 1
**Rating:** 2
**Confidence:** 3

**Summary:**

The authors provide an upper bound on the Rademacher complexity for transformer architecture by exploiting the Lipschitz properties of the model combined with the Talagrad contraction lemma. From the obtained bound, a data-dependent generalization bound is directly obtained. Furthermore, numerical experiments are provided to empirically validate the theory.

**Strengths:**

This paper contains an upper bound on the Rademacher complexity of a transformer architecture that includes the multihead attention mechanism and the layer normalization. The derivation of a generalization bound for transformers is an interesting direction that extends the known results on MLP to a broader class of models, which are widely adopted in practice and often forgotten by the more theoretical community.

**Weaknesses:**

The first point on which I have based my assessment is the originality of the work. The authors claim "We provide the first theoretical analysis of the self-attention mechanism within the Rademacher complexity framework" (lines 041-043); however, other works have been done in this direction. For instance, Trauger, Jacob, and Ambuj Tewari. "Sequence length independent norm-based generalization bounds for transformers." International Conference on Artificial Intelligence and Statistics. PMLR, 2024.

Furthermore, the overall writing of the paper could be improved. The results are not presented in rigorous form: some definitions and results used are not reported or are imprecise. The softmax activation function,  row-stochastic matrix (line 125), and the operation Concat (line 207) are not reported at all. The assumptions reported in the **Norm regime**  section are not stated clearly; for instance, the notion of weights $W$ has not been introduced yet. $X$ here denotes both in a single input and the space where of input matrix is, instead referring to uniform upper bounds. Given the theoretical nature of this paper, I would have appreciated a more rigorous and precise introduction and exposition of the work by using the math environments (definition, lemma, theorem, proofs, etc...), which would have increased the clarity and readability of the paper. The proof in Section 3.6 is unclear and not rigorous at all; even the statement is not clear. See, for instance, at line 321 the symbol $\approx$, which is not defined.

Regarding the experimental part, it seems that the only experiment shown is that the theoretical bound is monotonically increasing in $L$. However, it is not informative since the scaling of the theoretical bound is explicit, and the empirical generalization error is always zero, which does not correlate in any way to the theoretical bound. Furthermore, only one experiment is provided, which does not provide enough statistical evidence.

**Questions:**

I ask the authors the following questions:
1. How do you compare with "Sequence length independent norm-based generalization bounds for transformers." International    Conference on Artificial Intelligence and Statistics. PMLR, 2024"?
2. Why is the Rademacher complexity of the $\operatorname{Id}$ negligible? In general, it is not 0 (see Line 289).
3. Can you explain the exact statement that you prove in Section 3.6? What does the symbol $\approx$ mean precisely? And how do you use the norm stable assumption for deriving line 321 from 317? What is $SubLayer_L(Y)$ at line 332?
4. Can you explain what the experiments show? It seems that your theoretical bound does not relate at all to the empirical generalization bound.

**Details Of Ethics Concerns:**

.

---

### Note · Authors · 2026-07-16

**Comment:**

Not correct proofs.

**Withdrawal Confirmation:**

I have read and agree with the venue's withdrawal policy on behalf of myself and my co-authors.

---

### Meta-Review · Area_Chair_3T93 · 2026-01-03

**Summary:**

The paper studies theoretical generalization properties of Transformer architectures by deriving data-dependent bounds based on Rademacher complexity and a Lipschitz analysis of self-attention, leading to an end-to-end bound that scales with depth, sequence length, and weight norms, complemented by a small numerical check illustrating monotonic growth with depth. While the topic is relevant and the goal of unifying such dependencies is reasonable, the submission suffers from several major weaknesses that prevent acceptance. First, the originality and positioning are problematic: claims of being the first are inaccurate, and the related-work discussion omits or insufficiently contrasts multiple existing results on attention Lipschitzness and Transformer generalization, leaving the contribution unclear. Second, there are serious soundness and rigor issues, including missing or unstated assumptions, internal inconsistencies, undefined notation and operators, unclear approximations, and the absence of clearly stated theorems and proofs, which together undermine trust in the core results. Third, the presentation and professionalism fall short of the standards for a theory paper, with missing equation references, lack of formal structure, and vague or incomplete experimental descriptions that hinder clarity and reproducibility. Finally, the experimental validation is too weak to support the theory, as it is trivial, underspecified, and reports expected or uninformative behavior without meaningful connection to empirical generalization. A further concern is that the authors did not provide a rebuttal, which is a pity, as it removes the opportunity to clarify claims, address prior work, or resolve the substantive technical and presentation issues raised by all reviewers. Taken together, these points justify a rejection in the current form.

**Reviewer Concerns:**

No rebuttal.

**Reviewer Scores:**

No rebuttal.

---

### Decision · Program_Chairs · 2026-01-26

Reject